# Awake Testing during Deep Brain Stimulation Surgery Predicts Postoperative Stimulation Side Effect Thresholds

**DOI:** 10.3390/brainsci9020044

**Published:** 2019-02-18

**Authors:** Harrison C. Walker, Jesse Faulk, AKM Fazlur Rahman, Christopher L. Gonzalez, Patrick Roush, Arie Nakhmani, Jason L. Crowell, Barton L. Guthrie

**Affiliations:** 1Departments of Neurology and Biomedical Engineering, University of Alabama at Birmingham, Birmingham, AL 35294, USA; 2School of Medicine, University of Alabama at Birmingham, Birmingham, AL 35294, USA; jfaulk1@uabmc.edu (J.F.); roushw@health.missouri.edu (P.R.); 3Department of Biostatistics, University of Alabama at Birmingham, Birmingham, AL 35294, USA; frahman@uab.edu; 4Department of Neurology, University of Alabama at Birmingham, Birmingham, AL 35294, USA; clgonzalez@uabmc.edu (C.L.G.); jcrowell@uab.edu (J.L.C.); 5Department of Electrical and Computer Engineering, University of Alabama at Birmingham, Birmingham, AL 35294, USA; anry@uab.edu; 6Department of Neurosurgery, University of Alabama at Birmingham, Birmingham, AL 35294, USA; bguthrie@uabmc.edu

**Keywords:** awake behavioral testing, deep brain stimulation, movement disorders

## Abstract

Despite substantial experience with deep brain stimulation for movement disorders and recent interest in electrode targeting under general anesthesia, little is known about whether awake macrostimulation during electrode targeting predicts postoperative side effects from stimulation. We hypothesized that intraoperative awake macrostimulation with the newly implanted DBS lead predicts dose-limiting side effects during device activation in clinic. We reviewed 384 electrode implants for movement disorders, characterized the presence or absence of stimulus amplitude thresholds for dose-limiting DBS side effects during surgery, and measured their predictive value for side effects during device activation in clinic with odds ratios ±95% confidence intervals. We also estimated associations between voltage thresholds for side effects within participants. Intraoperative clinical response to macrostimulation led to adjustments in DBS electrode position during surgery in 37.5% of cases (31.0% adjustment of lead depth, 18.2% new trajectory, or 11.7% both). Within and across targets and disease states, dose-limiting stimulation side effects from the final electrode position in surgery predict postoperative side effects, and side effect thresholds in clinic occur at lower stimulus amplitudes versus those encountered in surgery. In conclusion, awake clinical testing during DBS targeting impacts surgical decision-making and predicts dose-limiting side effects during subsequent device activation.

## 1. Introduction

Deep brain stimulation (DBS) is effective for motor symptoms of Parkinson’s disease (PD), essential tremor, and dystonia that do not respond to medications and more conventional treatments [1,2,3,4,5]. Despite improvement at the group level, outcomes vary substantially in individuals, and surprisingly little is known about whether awake macrostimulation with the DBS electrode during surgery predicts postoperative stimulation side effects. The subthalamic nucleus (STN), globus pallidus pars interna (GPi), and ventral intermediate thalamus (VIM) each have distinct therapeutic properties and surrounding neuroanatomy [3,6]. As such, individual refinement of electrode positioning within these target regions plays an important role in optimizing clinical outcomes. Suboptimal lead placement remains an important problem that can lead to decreased efficacy, unwanted stimulation side effects, time-consuming postoperative programming sessions, and surgical revision to reposition the DBS electrode.

During DBS surgery, a linear array of electrode contacts is implanted in the brain, guided initially by anatomy from MR images. Although practices vary across centers, lead location is typically refined with single unit microelectrode recordings and/or macrostimulation with the DBS electrode [7,8,9]. Measurement of DBS side effect thresholds in awake patients allows mapping of the local functional neuroanatomy and the potential to tailor lead location in real time during surgery. Recent advances in intraoperative imaging have allowed targeting based on anatomy alone under general anesthesia without awake physiological and behavioral testing. Early experience with relatively small samples shows similar efficacy versus traditional awake surgery, although efficacy, adverse events, and postoperative side effects have not been compared in a prospective, randomized fashion [10,11,12,13,14,15].

Measuring whether awake macrostimulation predicts postoperative outcomes has the potential to inform decision-making in surgery and to clarify whether functional information might be lost with purely anatomical approaches. Our goal was to evaluate whether macrostimulation side effects during surgery (i.e., dysarthria, capsular, oculomotor, phosphenes, ataxia) across targets (STN, GPi, and VIM) predict similar dose-limiting side effects during device activation in clinic.

## 2. Materials and Methods

### 2.1. Participants

In this retrospective case series, we examined consecutive movement disorders patients who underwent DBS surgery at the University of Alabama at Birmingham between 1 January 2011 to 21 November 2016. Our routine practice is unilateral DBS surgery, followed by staged contralateral surgery, if and when it is needed [16,17]. Informed consent was not obtained individually as the data were acquired as part of routine care and entered into a de-identified, IRB-approved outcomes database.

### 2.2. Surgery and Initial Programming

One neurosurgeon (BLG) performed all surgeries using previously published methods [17]. On the day of surgery, we placed the stereotactic headframe, performed 1.5-T brain MRI, and then chose the initial target based on conventional stereotactic coordinates. We refined electrode targeting based upon individual anatomy, microelectrode recordings (in STN and GPi cases only), macrostimulation, and testing with the newly implanted Medtronic 3387 DBS lead (contacts spaced by 1.5 mm). Although we previously routinely performed microstimulation, over time our experience has been that microstimulation does not add additional information. We find macrostimulation with the newer Neuro Omega system to be more sensitive, and we now perform macrostimulation using the ring 3 mm above the microrecording electrode. All patients underwent awake behavioral testing with the DBS macroelectrode using contacts 3 anode and 0 cathode (3+ 0−), 60 µs pulse width, 160 Hz, constant voltage mode with increasing stimulus intensity in 0.5 V increments to ≥5 V, as tolerated. When indicated, additional testing was performed at other configurations (3+ 1−, 3+ 2−, etc.), and we refined the final electrode position, optimizing efficacy for motor symptoms versus acceptable side effects thresholds. Trained movement disorders specialists (nurse practitioners, neurologists) activated the DBS systems following a standardized survey approximately four weeks after implant [18,19,20].

### 2.3. Data Collection

Demographics, intra- and post-operative side effects with the associated voltage threshold, and DBS settings selected for chronic therapy were recorded. We categorized side effects as either motor (including dysarthria), sensory, cerebellar, autonomic, visual, oculomotor, or none, and all side effects were considered dose-limiting, except for “none” and transient contralateral sensory side effects. We also noted where macrostimulation led to adjustment of DBS electrode location in surgery (either a change in depth, a change in trajectory, or both). We only correlated side effect thresholds when results were available from both surgery and clinic at the same DBS electrode location and configuration.

### 2.4. Statistical Analyses

Analyses were performed in SAS version 9.4 and R. We calculated descriptive statistics (means, medians, standard deviations, interquartile ranges, and frequency distributions). Given that the mean stimulus intensity for therapy after initial postoperative programming with 3.5 ± 0.7 V, an intraoperative threshold of 3.5 V was selected for prediction of dose-limiting side effects in clinic. We performed multivariate logistic regression by employing generalized linear models with logit-link to estimate odds ratios ± 95% confidence intervals to quantify whether behavioral testing with DBS at 3+ 0− in surgery predicts the presence or absence of dose-limiting side effects in clinic across the different contacts on the DBS electrode array. A scatterplot matrix (R function “pairs’) was used to estimate whether side effect thresholds with 3+ 0− DBS during surgery correlate significantly with thresholds at 3+ 0−, 3+ 1−, case+ 0−, and case+ 1− in clinic. The LOWESS method (locally weighted scatter plot smoothing) was used to generate curve fits on the scatter plots. Bonferroni correction was applied to control for type I error for multiple simultaneous testing as there were 18 tests (3 targets and 6 contact pairs, odds ratios presented in Figure 1) and the critical *p*-value for these tests was set at 0.05/18 = 0.0028.

## 3. Results

### 3.1. Demographics

We evaluated 432 consecutive electrode placements, including 19 revisions—11 for prior infection (2.5%) and 8 for suboptimal efficacy and/or unacceptable side effect thresholds in clinic (1.9%). We excluded 48 lead placements because of incomplete data, either from surgery or clinic. Of the remaining 384 implants, there were 224 unilateral and 50 staged bilateral procedures in 273 unique patients. There were 203 lead implants in 170 PD patients, 79 in 71 ET patients, 36 in 26 dystonia patients, and 8 in 7 patients with various forms of cerebellar outflow tremor. The mean age at diagnosis and duration of disease were 53.5 ± 13.9 and 12.9 ± 7.8 years, respectively, and 43.6% of the surgeries were performed on women. At the first follow-up after initial programming, the mean voltage, pulse width, and frequency for chronic therapy were 3.5 ± 0.7 V, 70.5 ± 16.1 µs, and 155 ± 14 Hz, respectively.

Among 384 implants, electrode position was revised acutely in response to macrostimulation in 144 surgeries (37.5%). We adjusted depth within the same trajectory in 119 cases (31.0%), moved to a new trajectory in 70 (18.2%), and did both in 45 cases (11.7%). We excluded an additional 58 implants from the total 384 in whom electrode position was changed empirically during surgery without repeat threshold testing (typically raising the electrode dorsally by 2–3 mm within the same trajectory), leaving 326 implants.

### 3.2. Intraoperative Macrostimulation Predicts Postoperative DBS Side Effects

The primary results from these 326 lead placements show that overall intraoperative dose-limiting side effects with DBS at 3+ 0− at the final electrode position predicts whether dose-limiting side effects occur in clinic (χ^2^(1) = 9.59, *p* = 0.002). Odds ratios ± 95% confidence intervals show that intraoperative macrostimulation with DBS at 3+ 0− predicts whether postoperative dose-limiting side effects occur during monopolar stimulation across targets, diagnoses, and most DBS electrode configurations during device activation (Figure 1). Following Bonferroni correction for multiple comparisons, dose-limiting side effects from intraoperative macrostimulation still significantly predicts postoperative dose-limiting side effects for multiple DBS configurations across all targets. For the GPi target, macrostimulation in surgery at configuration 3+ 0− was predictive for postoperative side effects at <3.5 V in clinic at the more ventral contacts (contacts 0 and 1) but not the more dorsal contacts (contacts 2 and 3) in monopolar mode. Additionally, the absence of side effects during surgery with DBS at 3+ 0− at up to 5 V significantly predicts the absence of side effects at VIM (case+ 1−, *p* = 0.002; case+ 2−, *p* = 0.004) and GPi (case+ 2−, *p* = 0.002) in clinic. However, a lack of dose-limiting side effects in STN surgery did not predict the absence of side effects in clinic at our level of statistical power.

### 3.3. Thresholds for Intraoperative and Postoperative Dose-Limiting Side Effects

Typical side effect thresholds in clinic across subcortical targets are provided in Table 1. We also generated a correlation matrix across all targets and disease states when side effects were encountered in both the intraoperative and postoperative setting with the ‘pairs’ function from R, which demonstrates the relationship between intraoperative versus postoperative side effect thresholds at the following configurations: 3+ 0−, 3+ 1−, case+ 0−, and case+ 1− (Figure 2). Although there were significant correlations throughout, the strongest side effect correlations were noted between the intraoperative 3+ 0− configuration and postoperative monopolar thresholds at contacts 0 and 1.

## 4. Discussion

In this large consecutive series of patients with movement disorders, macrostimulation with bipolar DBS (3+ 0−) with the Medtronic 3387 lead during awake surgery (1) influences final electrode location in a substantial proportion of cases (37.5%) and (2) predicts dose-limiting side effects during device activation in clinic. Subgroup analyses show that macrostimulation in surgery predicts postoperative side effects across each brain target (STN, VIM, and GPi) and disease state (Figure 1). As a guide, we report postoperative side effect thresholds by brain target and active DBS contact, given the results of macrostimulation during targeting in surgery (Table 1). The overall results demonstrate that intraoperative macrostimulation informs clinical decision making, especially since DBS side effect thresholds are lower in surgery than those typically needed for efficacy in clinic.

Although macrostimulation during surgery predicts postoperative DBS side effects, the amplitude thresholds that elicit these side effects often differ considerably postoperatively during initial device activation (Figure 2). Importantly, these side effect thresholds are typically substantially lower in clinic versus in surgery across all stimulation configurations, targets, and disease states. Macrostimulation with DBS at 3+ 0− in surgery provided the best estimate for side effect amplitude thresholds with 3+ 1− stimulation in clinic versus other configurations. Further, correlations among intra- and post-operative thresholds were actually less robust than correlations between adjacent monopolar DBS contacts during subsequent device activation in clinic, highlighting some of the variability in the intraoperative testing results.

Several factors likely contribute to variability in DBS side effect thresholds over time. Patient anxiety, unfamiliarity with interpretation of stimulation effects, and time constraints might decrease the sensitivity or accuracy of behavioral testing during surgery. Although we use standard methods to elicit DBS side effects, clinicians may probe for side effects in slightly different ways to tailor therapy depending on individual patient response. Regarding lead positioning, we use fluoroscopy and, more recently, intraoperative CT to verify final electrode location. Nevertheless, slight shifts in electrode depth still may occur upon securing the skullcap or from brain shift (minimized in our case because these were exclusively unilateral procedures). Immediately after surgery, tissue impedances can decrease substantially, such that a larger current is typically delivered at a given voltage in clinic versus in surgery. This phenomenon is particularly consistent with our finding of lower side effect thresholds in clinic versus during surgery. Months to years after implant, tissue impedances tend to decline much more slowly, and the extent to which chronic changes in the electrode-tissue interface alter the side effect threshold and subjective perception is poorly understood [21,22].

Awake behavioral testing in surgery better predicts dose-limiting side effects for monopolar, as compared to bipolar stimulation mode during postoperative device activation (Figure 1). In contrast to VIM and GPi, dose-limiting side effects at the 3+ 0− pair in the STN region were more likely to be associated with similar side effects at the other contact pairings in clinic. Similarly, the absence of side effects during surgery in STN was less strongly predictive of absence of side effects in clinic versus the other targets. Together these findings illustrate the tendency for more frequent stimulation side effects and lower side effect thresholds in STN versus the other targets, presumably related to its smaller size and closer proximity to behaviorally relevant fibers of passage including the corticobulbar and corticospinal tracts [23,24].

This study has strengths and some potential limitations. Analyses from our large sample show significant correlations between intra- and post-operative side effect thresholds across multiple DBS targets and indications, in contrast to a prior study using a temporary macroelectrode with smaller dimensions (FHC) during bilateral STN surgery [25]. We performed behavioral testing in DBS surgery in the context of real world care rather than in the more restrictive inclusion/exclusion and time constraints of a prospective clinical trial. Despite this, we did not prospectively contrast whether different decision rules for adjusting electrode position in response to macrostimulation alters side effect thresholds or efficacy in clinic. Even very low side effect thresholds, particularly if restricted to monopolar configuration or to only a few electrode contacts, might not adversely affect outcomes in some cases. In contrast, avoiding side effects too aggressively during surgery could also compromise efficacy if the electrode is moved too far from the functional target. Although relevant to clinical decision-making, our findings might not apply as directly at centers with different practices for lead targeting/implant or for other DBS electrode geometries. Another potential limitation is that we did not explicitly identify the floor of the therapeutic window during intraoperative testing until more recently (a prior study suggested that therapeutic window is smaller postoperatively versus during surgery). Nevertheless, the ceiling of the therapeutic window is often limited by side effects, and low side effect thresholds are perhaps the most frequent indication for lead revisions.

DBS targeting under general anesthesia is an emerging technical approach that has been bolstered by recent advances in intraoperative imaging [26]. This practice improves patient comfort and increases access to DBS surgery, and retrospective data show similar efficacy versus traditional awake targeting (without advanced intraoperative imaging) [27]. Despite this, outcomes still vary considerably, and there are no prospective, class I studies to contrast different targeting approaches. Intraoperative physiology (single unit activity, local field potentials, macrostimulation) provides patient-specific information on anatomic boundaries and DBS efficacy/tolerability thresholds that can alter the final position of the DBS electrode significantly. Additionally, side effects from DBS (speech, gait) are poorly measured by the standard clinical rating instruments. Although a rigorous, prospective study is unlikely to occur, it remains possible that synergistic, complementary information from awake behavioral testing with modern intraoperative imaging better optimizes DBS efficacy/tolerability in some individuals versus a purely anatomic approach.

Although awake macrostimulation during DBS surgery commonly influences lead location and predicts postoperative side effects, biomarkers are needed to understand and predict interactions between DBS and motor and non-motor sub-circuits in the stimulated region. These biomarkers could be used to advance increasingly complex device technologies such as directional and closed loop stimulation. Future studies should identify such biomarkers and prospectively contrast efficacy, side effects, cost, and surgical adverse events following DBS targeting while awake versus under general anesthesia.

## 5. Conclusions

Awake clinical testing during DBS targeting commonly impacts surgical decision-making and predicts dose-limiting side effects during subsequent device activation in clinic.

## Figures and Tables

**Figure 1 brainsci-09-00044-f001:**
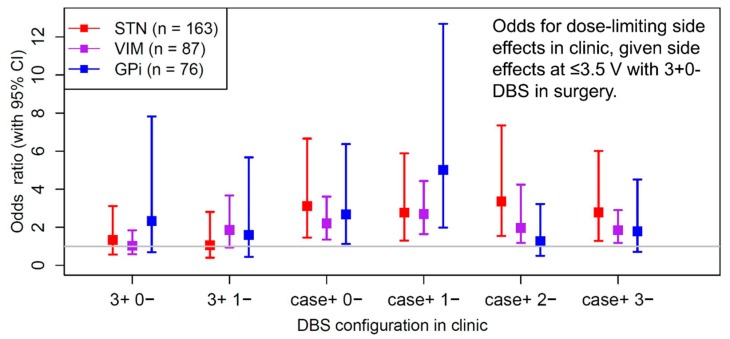
Means, standard deviations, and quartiles for dose-limiting side effect thresholds in clinic by DBS target, given dose-limiting side effects at <3.5 V, ≥3.5 V, or no dose-limiting side effects in surgery. Proportions reflect number of patients with dose-limiting side effects in surgery by target and in clinic within each category.

**Figure 2 brainsci-09-00044-f002:**
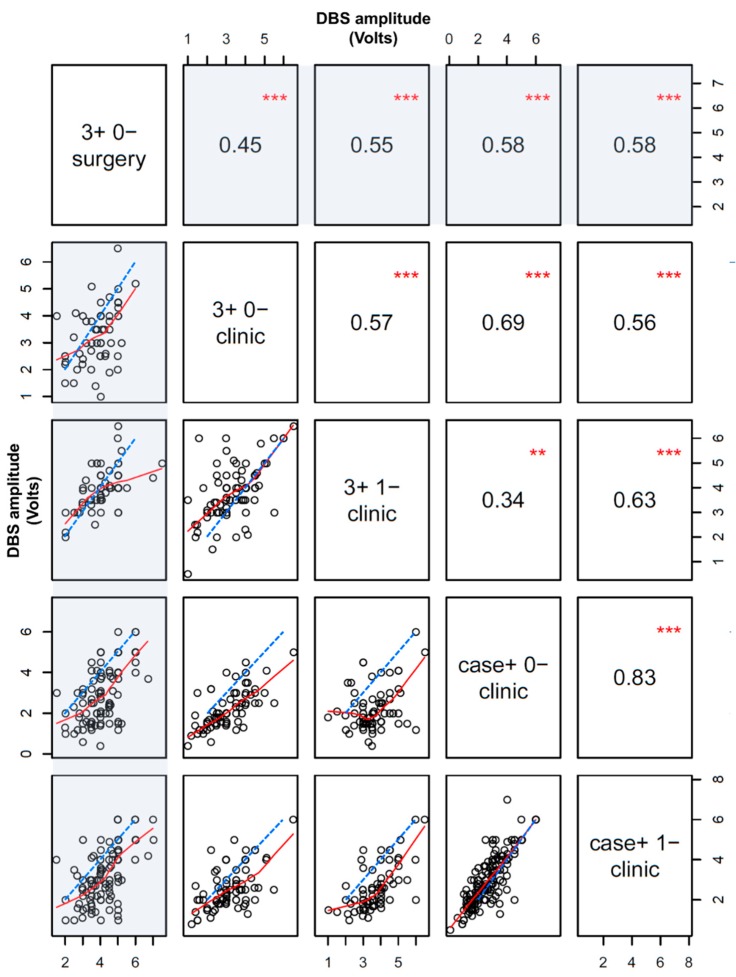
Scatter plot matrix for dose-limiting side effect thresholds (in Volts) during DBS surgery versus in clinic within individuals. Points are only displayed and analyzed if a dose-limiting side effect was detected in both surgery and clinic. Red lines are a fit based on the LOWESS correlation method, and dashed blue lines are the unity slope line (if the intra- and post-operative side effect thresholds were identical). The first column and row in the matrix plot and measure the correlation for intraoperative versus postoperative side effect thresholds at various DBS configurations (shaded). Non-shaded areas display postoperative correlations only. Absolute correlation coefficients and statistical significance are displayed in corresponding opposite panels on the matrix (*** *p* < 0.001, ** *p* < 0.01).

**Table 1 brainsci-09-00044-t001:** DBS side effect thresholds in clinic, by stereotactic target (mean ± SD, proportion, and inter-quartile range).

	Threshold with 3+ 0− DBS in Surgery <3.5			Threshold in Surgery ≥3.5 V			No Dose-Limiting Side Effect in Surgery		
Target	STN (*n* = 124)	VIM (*n* = 59)	GPi (*n* = 75)	STN (*n* = 124)	VIM (*n* = 59)	GPi (*n* = 75)	STN (*n* = 124)	VIM (*n* = 59)	GPi (*n* = 75)
**(%)**	21 (17%)	12 (20%)	2 (4%)	59 (48%)	32 (54%)	17 (21%)	44 (35%)	15 (25%)	56 (75%)
**case+ 0−**	1.8 ± 0.8 (0.86)	1.9 ± 0.7 (0.67)	3.1 ± 0.1 (1.0)	2.4 ± 0.9 (0.64)	2.7 ± 0.9 (0.56)	2.7 ± 1.1 (0.94)	2.8 ± 1.0 (0.66)	2.6 ± 0.7 (0.67)	3.6 ± 1.3 (0.43)
	1.4, 1.6, 2.0	1.4, 1.8, 2.5	3.1, 3.1, 3.5	1.7, 2.4, 2.9	2.2, 2.8, 3.2	2.2, 2.9, 3.3	2.0, 2.8, 3.5	2.3, 2.6, 3.0	3.0, 3.5, 4.4
**case+ 1−**	2.1 ± 0.8 (0.81)	2.4 ± 0.7 (0.58)	4.1 ± 1.3 (1.0)	2.5 ± 0.8 (0.68)	2.9 ± 1.1 (0.63)	3.1 ± 0.9 (0.71)	2.9 ± 0.9 (0.64)	3.5 ± 0.9 (0.80)	4.1 ± 1.5 (0.46)
	1.5, 2.0, 2.5	1.8, 2.5, 2.8	3.7, 4.1, 4.6	2.0, 2.5, 3.0	2.2, 2.8, 3.7	2.2, 3.4, 3.8	2.0, 2.9, 3.7	2.9, 3.9, 4.0	3.5, 4.2, 4.7
**case+ 2−**	2.5 ± 0.8 (0.81)	2.6 ± 0.9 (0.67)	4.1 (0.50)	2.7 ± 0.9 (0.69)	3.3 ± 1.3 (0.59)	3.0 ± 1.0 (0.47)	3.0 ± 0.8 (0.61)	4.1 ± 1.2 (0.60)	3.7 ± 1.1 (0.36)
	1.9, 2.3, 2.6	1.8, 2.4, 3.0	4.1, 4.1, 4.1	2.0, 2.5, 3.4	2.5, 3.0, 4.0	2.4, 3.0, 3.8	2.4, 3.0, 3.6	3.7, 4.0, 4.5	3.4, 3.8, 4.1
**case+ 3−**	2.7 ± 1.4 (0.67)	3.3 ± 1.0 (0.58)	0.0	3.0 ± 0.9 (0.46)	3.1 ± 1.6 (0.38)	3.1 ± 1.2 (0.41)	3.4 ± 0.8 (0.34)	4.4 ± 2.0 (0.40)	3.8 ± 1.4 (0.30)
	1.5, 2.6, 3.6	2.8, 3.5, 3.8		2.5, 3.0, 3.5	2.0, 3.1, 4.0	2.7, 3.5, 3.8	3.0, 3.2, 4.0	3.2, 3.9, 4.9	3.0, 3.6, 4.5
**3+ 0−**	2.6 ± 0.9 (0.52)	2.7 ± 0.9 (0.58)	0.0	3.1 ± 1.0 (0.36)	3.1 ± 1.1 (0.28)	2.5 ± 0.9 (0.41)	2.7 ± 0.8 (0.22)	3.9 ± 1.0 (0.40)	3.6 ± 0.9 (0.16)
	2.2, 2.4, 3.1	2.0, 2.5, 3.2		2.5, 3.0, 3.8	3.0, 3.0, 3.5	2.2, 2.7, 3.2	2.2, 2.8, 3.0	3.5, 3.8, 4.4	3.5, 4.0, 4.0
**3+ 1−**	3.0 ± 1.0 (0.48)	3.4 ± 0.8 (0.58)	0.0	3.8 ± 0.9 (0.31)	4.3 ± 0.7 (0.31)	4.0 ± 0.9 (0.41)	3.2 ± 1.0 (0.18)	3.6 ± 1.4 (0.27)	3.0 ± 1.2 (0.09)
	2.4, 3.2, 3.5	3.1, 3.5, 4.0		3.1, 3.9, 4.2	4.0, 4.3, 5.0	3.4, 3.5, 4.5	2.9, 3.1, 3.5	2.8, 3.8, 4.6	2.1, 2.9, 3.0

Odds ratios ±95% confidence interval for postoperative dose-limiting side effects, given that dose-limiting side effects occurred during awake behavioral testing with DBS at 3+ 0− in surgery at ≤3.5 V. DBS: deep brain stimulation; STN: subthalamic nucleus; VIM: ventral intermediate thalamus; GPi: globus pallidus pars interna.

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
