# Peer review of "Awake Testing during Deep Brain Stimulation Surgery Predicts Postoperative Stimulation Side Effect Thresholds"

_brainsci, 2019, doi:10.3390/brainsci9020044_

Round 1
Reviewer 1 Report
This is a timely and extremely well-done analysis that shows that intraoperative side effects with macrostimulation predict post-operative side effects. This study is likely to be highly informative of the debate on awake vs. asleep DBS.
I have extremely minor comments:
* I might suggest clarifying that this is a retrospective consecutive case series
* Line language in line 85 is somewhat colloquial - I might revise the rationale for 'moved away from'
* I wasn't sure what the x/y axis labels were on Figure 2 - they may be volts
Author Response
Reviewer 1:
This is a timely and extremely well-done analysis that shows that intraoperative side effects with macrostimulation predict post-operative side effects. This study is likely to be highly informative of the debate on awake vs. asleep DBS.
I have extremely minor comments:
* I might suggest clarifying that this is a retrospective consecutive case series
Thank you for identifying this omission. We added this clarification in line 74.
* Line language in line 85 is somewhat colloquial - I might revise the rationale for 'moved away from'
We agree that this phrasing needed improvement and clarification. At our institution, we previously routinely performed microstimulation, but over time we found that this did not add new information; rather, we find macrostimulation (using the ring contact 3 mm above the microrecording electrode) to be more sensitive, and thus this is now our routine practice. We added this explanation in lines 86-89.
* I wasn't sure what the x/y axis labels were on Figure 2 - they may be volts
We agree; Figure 2 did not clearly identify the units (volts). We updated Figure 2 to include the units on both the X and Y axes.
Reviewer 2 Report
This paper correlates side effects in vs. out of the OR, and determines that there is a relationship, which has never been done. There are quite a few patients here and advanced statistical methods, which supports this analysis. The text is clear and easy to read. The conclusions consistent with the evidence and arguments presented. Authors addresses the main question posed.
Nice paper, well written, lots of data lumped together, but there is nothing unexpected or really novel to be said, here, especially now with the newer lead the whole landscape is changing and thus clinical relevance, especially for the near future is limited.
Author Response
Reviewer 2:
This paper correlates side effects in vs. out of the OR, and determines that there is a relationship, which has never been done. There are quite a few patients here and advanced statistical methods, which supports this analysis. The text is clear and easy to read. The conclusions consistent with the evidence and arguments presented. Authors addresses the main question posed.
Nice paper, well written, lots of data lumped together, but there is nothing unexpected or really novel to be said, here, especially now with the newer lead the whole landscape is changing and thus clinical relevance, especially for the near future is limited.
Thank you for your time and effort in reviewing our paper. We agree that the landscape is indeed changing, and we respectfully submit that our data add new information to the conversation in the growing debate between whether to perform DBS targeting in asleep versus awake patients—specifically, that intraoperative macrostimulation in the awake patient has predictive value for postoperative programming.
Reviewer 3 Report
This manuscript is very well written and conclusive. It is of interest for clinicians that perform DBS surgery, indicating that intraoperative macrostimulation is useful to estimate postoperative side effects. The statistical analysis is elaborated and has sufficient power. It might be obvious that intraoperative stimulation effects correlate with postoperative side effects, however the authors provide a statistical fundament which may inform clinicians practice when debating awake DBS/general anesthesia. This might be even more relevant with newer devices that provide directional leads.
I have only a few minor comments and annotations:
1. Line 145: For the GPi target, macrostimulation in surgery at configuration 3+0- not predictive for postoperative side effects at<3.5V in clinic at the more dorsal contacts (contacts 2 and 3) in monopolar mode. Please rephrase
2. In Figure 1, what is the purpose of indicating panel “A”? Furthermore, the authors should explain the grey line – I assume it represents Bonferroni correction?
3. Figure 2 is quite confusing, I have to admit. I think I qould be much easier to understand the figure if significances are written in the corresponding plot/panel.
4. How to the authors interpret their finding that the association between 3+0- setting at surgery and c+1-/c+0- setting postoperatively is the strongest and not with 3+0- postoperatively?
5. Line 212ff: “In contrast to VIM and GPi, dose-limiting side effects at the 3+0- pair in the STN region were more likely to be associated with similar side effects at the other contact pairings in clinic. Similarly, the absence of side effects during surgery in STN was less strongly predictive of absence of side effects in clinic versus the other targets. Together these findings illustrate the tendency for more frequent stimulation side effects and lower side effect thresholds in STN versus the other targets”
This conclusion is hard to follow. If there is no side effect during surgery, I would conclude that lead positioning is good and fewer side effects should occur postoperatively, even more so if the target is small. Can the authors elaborate on this difference between targets?
Author Response
Reviewer 3:
This manuscript is very well written and conclusive. It is of interest for clinicians that perform DBS surgery, indicating that intraoperative macrostimulation is useful to estimate postoperative side effects. The statistical analysis is elaborated and has sufficient power. It might be obvious that intraoperative stimulation effects correlate with postoperative side effects, however the authors provide a statistical fundament which may inform clinicians practice when debating awake DBS/general anesthesia. This might be even more relevant with newer devices that provide directional leads.
I have only a few minor comments and annotations:
1. Line 145: For the GPi target, macrostimulation in surgery at configuration 3+0- not predictive for postoperative side effects at<3.5V in clinic at the more dorsal contacts (contacts 2 and 3) in monopolar mode. Please rephrase
We agree that the phrasing of this result could be improved. We found that, for the GPi target, macrostimulation in surgery was predictive for postoperative side effects in clinic at the more ventral contacts (contacts 0, 1) but not predictive at the more dorsal contacts (2, 3). This is best seen in Figure 1. We added this clarification in lines 148-150.
2. In Figure 1, what is the purpose of indicating panel “A”? Furthermore, the authors should explain the grey line – I assume it represents Bonferroni correction?
Do you have a way to remove this from the figure?
Thank you for pointing this out; the indication of panel “A” was superfluous and has been removed in our updated figure. The grey line indicates an odds ratio of 1.
3. Figure 2 is quite confusing, I have to admit. I think I qould be much easier to understand the figure if significances are written in the corresponding plot/panel.
We agree that the data in Figure 2 are complex, and we hope our modifications to Figure 2 make the data easier to understand. We shaded the first column and the first row, which together plot and measure the correlation for intraoperative versus postoperative side effect thresholds at various DBS configurations. The non-shaded areas display postoperative correlations only.
4. How to the authors interpret their finding that the association between 3+0- setting at surgery and c+1-/c+0- setting postoperatively is the strongest and not with 3+0- postoperatively?
We agree this is a very interesting question. It may be a manifestation of transient edema / changes in local tissues interface intraoperatively versus 1 month post-op. Also from the standpoint of statistics, the stronger correlations with postoperative monopolar configuration may arise because monopolar mode is more potent and likely to elicit side effects postoperatively. So across our entire sample in Fig 2 column 1, there are more pairs of values (dots) for intraoperative bipolar to postoperative monopolar mode correlations, versus intraoperative bipolar to postoperative bipolar correlations. Interpreted together, Figs 1 and 2 suggest that intraoperative bipolar mode side may or may not be associated with postoperative bipolar side effects, but if there are side effects in both venues, the stimulus amplitude at which the side effects occur is correlated.
5. Line 212ff: “In contrast to VIM and GPi, dose-limiting side effects at the 3+0- pair in the STN region were more likely to be associated with similar side effects at the other contact pairings in clinic. Similarly, the absence of side effects during surgery in STN was less strongly predictive of absence of side effects in clinic versus the other targets. Together these findings illustrate the tendency for more frequent stimulation side effects and lower side effect thresholds in STN versus the other targets”
This conclusion is hard to follow. If there is no side effect during surgery, I would conclude that lead positioning is good and fewer side effects should occur postoperatively, even more so if the target is small. Can the authors elaborate on this difference between targets?
Thank you for this comment; we agree that we could improve clarity in this statement. The STN target was associated with more frequent postoperative side effects versus the GPi and VIM targets. Any predictive assay has the possibility for false negative or false positive results. Given the greater prior probability for side effects in STN, intraoperative testing in this target may result in more false negative tests versus the other targets, in which the prior probability of post-op side effects is lower.